# Lys417 acts as a molecular switch that regulates the conformation of SARS-CoV-2 spike protein

Qibin Geng[1,2†], Yushun Wan[1,2†], Fu-Chun Hsueh[1,2], Jian Shang[1,2], Gang Ye[1,2], Fan Bu[1,2], Morgan Herbst[1,2], Rowan Wilkens[1,2], Bin Liu[3]*, Fang Li[1,2]*

[1]Department of Pharmacology, University of Minnesota Medical School, Minneapolis, United States; [2]Center for Coronavirus Research, University of Minnesota, Minneapolis, United States; [3]Hormel Institute, University of Minnesota, Austin, United States

**Abstract** SARS-CoV-2 spike protein plays a key role in mediating viral entry and inducing host immune responses. It can adopt either an open or closed conformation based on the position of its receptor-binding domain (RBD). It is yet unclear what causes these conformational changes or how they influence the spike's functions. Here, we show that Lys417 in the RBD plays dual roles in the spike's structure: it stabilizes the closed conformation of the trimeric spike by mediating inter-spike–subunit interactions; it also directly interacts with ACE2 receptor. Hence, a K417V mutation has opposing effects on the spike's function: it opens up the spike for better ACE2 binding while weakening the RBD's direct binding to ACE2. The net outcomes of this mutation are to allow the spike to bind ACE2 with higher probability and mediate viral entry more efficiently, but become more exposed to neutralizing antibodies. Given that residue 417 has been a viral mutational hotspot, SARS-CoV-2 may have been evolving to strike a balance between infection potency and immune evasion, contributing to its pandemic spread.

*For correspondence:
liu00794@umn.edu (BL);
lifang@umn.edu (FL)

†These authors contributed equally to this work

Competing interest: The authors declare that no competing interests exist.

## Editor's evaluation

In this valuable study, Li et al. identify a molecular switch within the SARS-CoV-1/2 spike proteins, in position 417. Based on comparisons between CoV-1 and CoV-2 sequences, the authors propose that the nature of the residue of this position determines whether the spike favors a closed or an open conformation. The authors present solid cryo-EM and biochemical data in support of their hypothesis. The paper will be of general interest to those interested in molecular mechanisms of viral infection by SARS-CoV-2 and other coronaviruses.

## Introduction

Coronaviruses have long infected humans, yet none caused the same devastation as SARS-CoV-2 has (*Li et al., 2020*; *Huang et al., 2020*). For instance, a virulent and lethal coronavirus, SARS-CoV-1, caused a much smaller outbreak in 2002–2003 (*Lee et al., 2003*; *Peiris et al., 2003*). Numerous human coronaviruses such as NL63-CoV cause common colds annually (*Fouchier et al., 2004*; *van der Hoek et al., 2004*). With an intermediate virulence, SARS-CoV-2 causes a fatality rate that is significantly lower than that of SARS-CoV-1, but much higher than that of NL63-CoV. Because of its intermediate virulence, SARS-CoV-2 carriers show clinical signs that facilitate the spread of the virus: they may develop mild or no symptoms, experience delayed onset of symptoms, develop low levels of neutralizing antibodies, or endure prolonged virus shedding period (*Wu et al., 2020*; *Zhou et al.,*

*2020*; *Wölfel et al., 2020*; *Gao et al., 2020*; *Kronbichler et al., 2020*). These features contribute to the wide spread of SARS-CoV-2 and severe health outcomes, triggering a global COVID-19 pandemic that was unprecedented in the era of modern medicine. Understanding the molecular determinants of COVID-19 is crucial for comprehending the recent pandemic and preventing future ones.

SARS-CoV-2 spike protein played a central role in the COVID-19 pandemic. It guides viral entry into host cells and is also a major target for the host immune responses (*Du et al., 2009*; *Li, 2016*). On newly packaged virus particles, the trimeric spike protein has a pre-fusion structure in which three receptor-binding S1 subunits sit on top of a trimeric membrane-fusion S2 stalk (*Figure 1A*). During viral entry, a receptor-binding domain (RBD) in S1 binds to a receptor on host cell surface for viral attachment and S2 switches to a post-fusion structure for the fusion of viral and host membranes (*Li, 2016*; *Li, 2015*). SARS-CoV-2, SARS-CoV-1, and NL63-CoV can all use ACE2 as their receptor (*Li et al., 2003*; *Li et al., 2005*; *Shang et al., 2020c*; *Wan et al., 2020a*; *Wu et al., 2009*). Early in the pandemic, we described three distinct characteristics of SARS-CoV-2 spike: its exceptional binding affinity for human ACE2, its proteolytic activation by human protease furin, and its existence as a mixture of two conformations: an open conformation in which the RBD is exposed and accessible to ACE2 and a closed conformation in which the RBD is buried and inaccessible to ACE2 (*Shang et al., 2020b*). While much has been learned about the first two characteristics (*Shang et al., 2020c*; *Lan et al., 2020*; *Essalmani et al., 2022*; *Peacock et al., 2021*; *Geng et al., 2022*; *Zhang et al., 2023*), the third still remains poorly understood. Interestingly, SARS-CoV-1 spike is predominantly in the open conformation, whereas NL63-CoV spike remains closed (*Walls et al., 2016*; *Wrobel et al., 2020*; *Gui et al., 2017*). Thus, two key questions arise: what molecular switches regulate the conformation of coronavirus spikes and how do the conformational changes influence coronaviruses' infectivity and host immune responses?

In this study, we used cryo-EM and biochemical approaches to identify spike residue 417 as a molecular switch that regulates the conformation of SARS-CoV-2 spike. We delved deeper into how this molecular switch affects receptor binding, viral entry, and immune evasion by SARS-CoV-2. Through regulation of its spike's conformations, SARS-CoV-2 may have struck a balance between infection potency and immune evasiveness.

## Results

To understand the molecular mechanism that controls the spike's switching between open and closed conformations, we compared the sequences of SARS-CoV-2 and SARS-CoV-1 spikes in the context of their tertiary structures. We identified residue 417 as potentially a key difference between the two spikes: in the closed SARS-CoV-2 trimeric spike, Lys417 in the RBD from one spike subunit forms a hydrogen bond with the main chain of Asn370 in the RBD from another spike subunit, stabilizing the RBDs in the closed conformation (*Walls et al., 2020*; *Figure 1B*). In SARS-CoV-1 spike, however, this residue becomes a valine (*Li et al., 2005*; *Shang et al., 2020a*; *Figure 1C*), losing its capability to form the hydrogen bond with another spike subunit and potentially destabilizing the closed conformation of the spike. Thus, we hypothesized that a K417V mutation causes more SARS-CoV-2 spike molecules to take the open conformation. To test this hypothesis, we investigated the impact of the K417V mutation on the structure and function of SARS-CoV-2 spike using a combination of cryo-EM and biochemical approaches.

To examine how the K417V mutation affects the conformation of SARS-CoV-2 spike, we performed cryo-EM analysis of the RBD conformation of SARS-CoV-2 spike ectodomain containing either Lys417 or Val417. To this end, we introduced the K417V mutation into SARS-CoV-2 spike ectodomain and purified both versions of the protein: K417-spike containing Lys417 and V417-spike containing Val417. Due to the lack of the transmembrane anchor, the recombinant spike ectodomain was intrinsically unstable. To stabilize the recombinant spike ectodomain, we introduced six proline mutations into its S2 subunit to lock up its pre-fusion structure, introduced mutations to its furin motif to prevent it from being cleaved during molecular maturation, and added a C-terminal foldon tag to facilitate its trimerization (*Shang et al., 2020b*; *Hsieh et al., 2020*; *Ye et al., 2022*). We successfully obtained the stabilized K417-spike and V417-spike and collected cryo-EM data on both proteins (*Figure 2—figure supplements 1 and 2*; *Supplementary file 1*). For the K417-spike, 55% of the molecules were in the closed conformation with all three RBDs down, whereas the rest of the molecules (45%) were in the open conformation with one RBD up and two RBDs down (*Figure 2A and B*). In contrast, for the

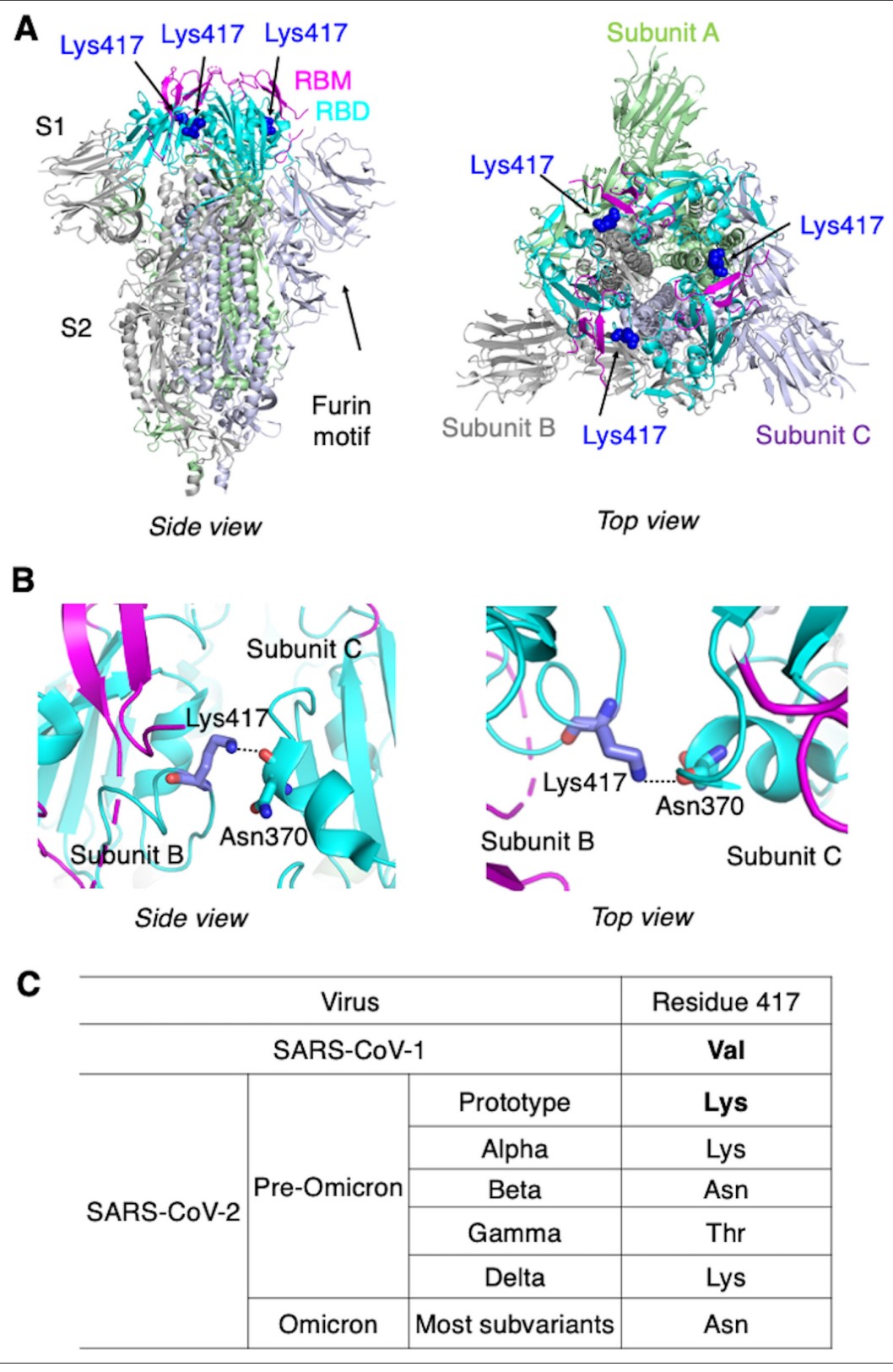

**Figure 1.** Identification of residue 417 as a molecular switch that regulates the conformation of SARS-CoV-2 spike. (**A**) Structure of trimeric SARS-CoV-2 spike ectodomain in the closed conformation with three receptor-binding domains (RBDs) down (PDB 6VXX). Each monomeric subunit of the spike trimer is colored differently. The RBD contains a core structure (in cyan) and a receptor-binding motif (RBM; in magenta). Lys417 in the RBD is shown as

*Figure 1 continued on next page*

*Figure 1 continued*

blue sticks. (**B**) A hydrogen bond is formed between the side chain of Lys417 from one spike subunit and the main chain of Asn370 from another spike subunit, stabilizing the trimeric spike in the closed conformation. (**C**) Residue 417 is a valine in SARS-CoV-1 spike and has been a mutational hotspot in later SARS-CoV-2 variants.

V417-spike, these numbers became 32 and 68% for the closed and open conformations, respectively (*Figure 2C and D*). Therefore, the K417V mutation allowed more SARS-CoV-2 spike molecules to take the open conformation, confirming our hypothesis.

While the above cryo-EM approach examined the conformation of recombinant spike ectodomain, we further examined the impact of the K417V mutation on the membrane-anchored and full-length SARS-CoV-2 spike that do not contain S2 or furin motif mutations. It has been shown that ACE2 only binds to the standing-up RBD in the open spike (*Gui et al., 2017*; *Ye et al., 2021*), suggesting that the open spike conformation is necessary for ACE2 binding. Hence, we characterized the capabilities of K417-spike and V417-spike in binding ACE2. First, we expressed the K417-spike and V417-spike on human cell surface and performed a protein pull-down assay using recombinant human ACE2 as

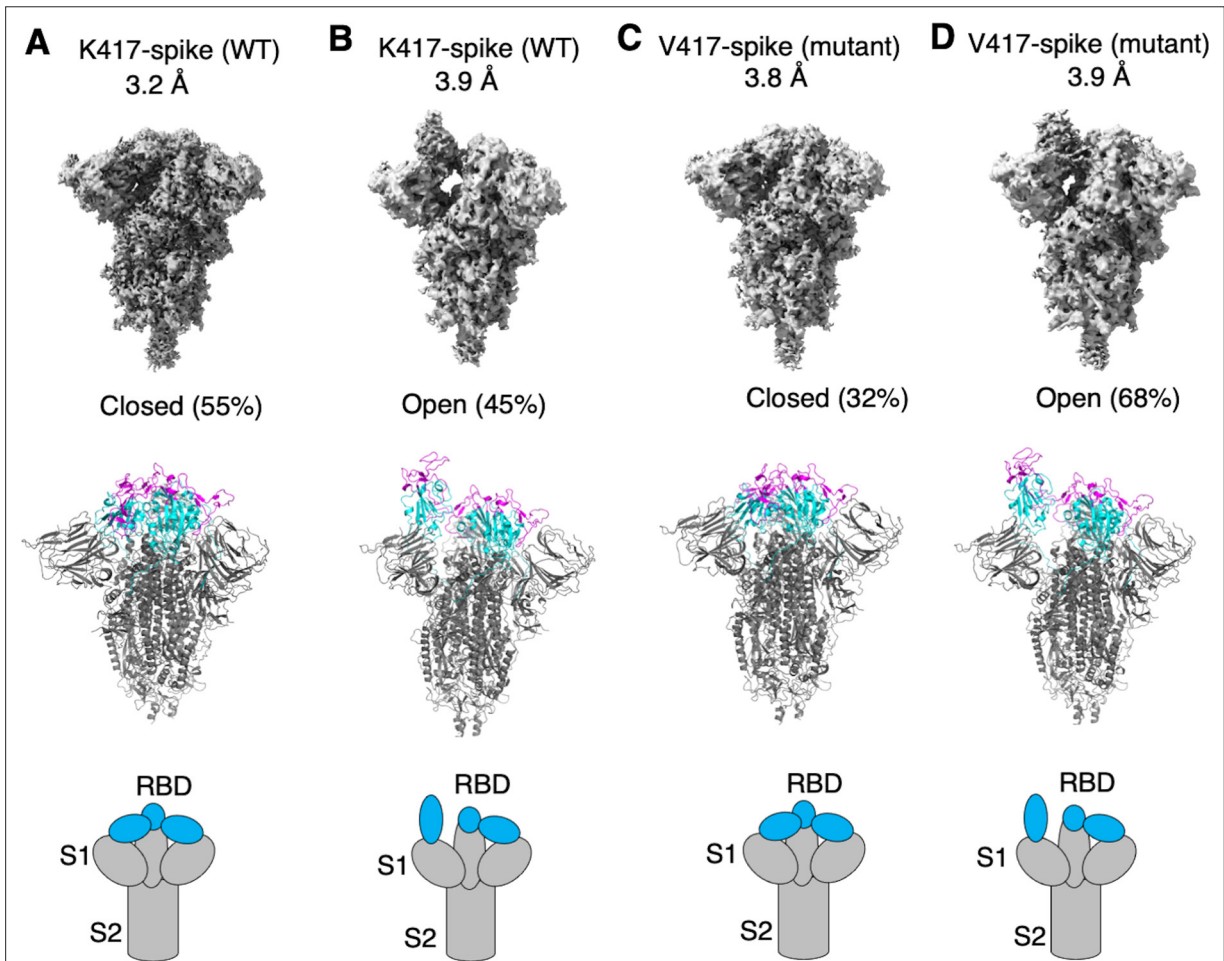

**Figure 2.** Cryo-EM analyses of residue 417 in regulating the conformation of recombinant SARS-CoV-2 spike ectodomain. Prototypic SARS-CoV-2 spike ectodomains containing either Lys417 (as in prototypic SARS-CoV-2) or Val417 (as in SARS-CoV-1) were subjected to cryo-EM analyses and the particle distributions for open and closed conformations were calculated. Cryo-EM densities (top), atomic models (middle), and schematic presentations (bottom) for each of the protein classes are shown. (**A**) K417-spike in the closed conformation. (**B**) K417-spike in the open conformation. (**C**) V417-spike in the closed conformation. (**D**) V417-spike in the open conformation.

The online version of this article includes the following figure supplement(s) for figure 2:

**Figure supplement 1.** Flow chart of cryo-EM data processing for prototypic SARS-CoV-2 spike ectodomain containing Lys417 (K417-spike).

**Figure supplement 2.** Flow chart of cryo-EM data processing for prototypic SARS-CoV-2 spike ectodomain containing Val417 (V417-spike).

the bait and the cell-surface-anchored spikes as the target (*Figure 3A*). For cross-validation, both His-tagged ACE2 and Fc-tagged ACE2 were used. We previously showed that this pull-down assay is a reliable method to probe the RBD conformation of cell-surface-anchored spikes, with higher pull-down levels of spikes associated with more spike molecules in the open conformation (*Shang et al., 2020b*). Western blot analysis showed that some of the spike molecules had been cleaved at the S1/S2 boundary, confirming the intact furin motif (*Figure 3A*). The pull-down result showed that, compared to K417-spike, more V417-spike molecules had been pulled down by ACE2, suggesting that V417-spike had an increased probability of binding to ACE2 (*Figure 3A*). Second, we conducted a flow cytometry assay to detect the interactions between recombinant human ACE2 and cell-surface-anchored spikes (*Figure 3B*, *Figure 3—figure supplement 1*). Here, two versions of both K417-spike and V417-spike were used: one containing the intact furin motif and the other the inactivated furin motif. The result showed that V417-spike binds to ACE2 better than does K417-spike, no matter whether the furin motif was intact or inactivated (*Figure 3B*, *Figure 3—figure supplement 1*). Taken together, both protein pull-down and flow cytometry results reveal that the K417V mutation allowed more SARS-CoV-2 spike molecules to take the open conformation, confirming the cryo-EM data and our hypothesis.

To understand the role of the spike's conformations in viral entry, we conducted a pseudovirus entry assay. Specifically, retroviruses pseudotyped with SARS-CoV-2 spike (i.e., SARS-CoV-2 pseudoviruses) were evaluated for their capability to enter ACE2-expressing human cells. Two types of SARS-CoV-2 pseudoviruses were used: the K417-spike-charged pseudoviruses (i.e., K417-pseudoviruses) and V417-spike-charged pseudoviruses (V417-pseudoviruses). Both types of pseudoviruses contain the furin motif in their spike. Western blot analysis showed that some of the spike molecules had been cleaved at the S1/S2 boundary when expressed on pseudovirus surface, confirming the intact furin motif (*Figure 3C*). The pseudovirus entry result showed that V417-pseudoviruses entered cells more efficiently than did K417-pseudoviruses (*Figure 3C*). Thus, the open spike enhances not only ACE2 binding, but also viral entry.

To understand the role of the spike's conformation in immune evasion, we analyzed how neutralizing antibodies recognize the RBD in the context of the trimeric spike protein. To this end, we summarized the binding sites of known neutralizing antibodies on the RBD (*Figure 4A*). These neutralizing antibodies were discovered individually from COVID-19 patients. They generally recognize five groups of neutralizing epitopes on the RBD (*Starr et al., 2021*; *Dussupt et al., 2021*; *Scheid et al., 2021*; *Fedry et al., 2021*). Three groups of these RBD epitopes are only accessible to neutralizing antibodies when the spike adopts the open conformation (*Starr et al., 2021*; *Dussupt et al., 2021*). In comparison, the other two groups of RBD epitopes are accessible to neutralizing antibodies when the spike adopts either open or closed conformation (*Scheid et al., 2021*; *Fedry et al., 2021*). In addition to the above conventional antibodies, neutralizing nanobodies (which are single-domain antibodies derived from camelid animals) have also been discovered to target the RBD. Because of their single-domain structure, nanobodies are known to bind to cryptic epitopes on viral targets. We recently discovered four RBD-targeting nanobodies, named Nanosota-1, -2, -3, and -4 (*Ye et al., 2021*; *Ye et al., 2023*; *Figure 4B*). Among them, Nanosota-2 binds to an epitope on the RBD that almost completely overlaps with the ACE2-binding site. Interestingly, the Nanosota-2 epitope is only accessible to Nanosota-2 when the RBD is in the open conformation (*Ye et al., 2023*). This result suggests that even with the small size of nanobodies, SARS-CoV-2 may be able to evade some of them when its spike is in the closed conformation. Overall, these epitope-based analyses reveal that SARS-CoV-2 spike in the closed conformation can evade significant amounts of neutralizing antibodies, confirming our previous hypothesis that the closed conformation of the spike is a viral strategy for immune evasion (*Shang et al., 2020b*).

To further understand the role of the K417 mutation in ACE2 binding and viral entry, we examined how the K417 mutation directly affects the RBD's binding affinity for ACE2 instead of the trimeric spike's. The structure of SARS-CoV-2 RBD complexed with human ACE2 showed that Lys417 in SARS-CoV-2 RBD is directly involved in ACE2 binding by forming a salt bridge with Asp30 in human ACE2 (*Lan et al., 2020*; *Figure 5A*). In contrast, the corresponding residue in SARS-CoV-1 RBD is a valine that does not form any direct interactions with human ACE2 (*Li et al., 2005*; *Figure 5B*). Here, we introduced the K417V mutation to SARS-CoV-2 RBD and investigated the binding interactions between the recombinant SARS-CoV-2 RBD (containing either Lys417 or Val417) and cell-surface-anchored

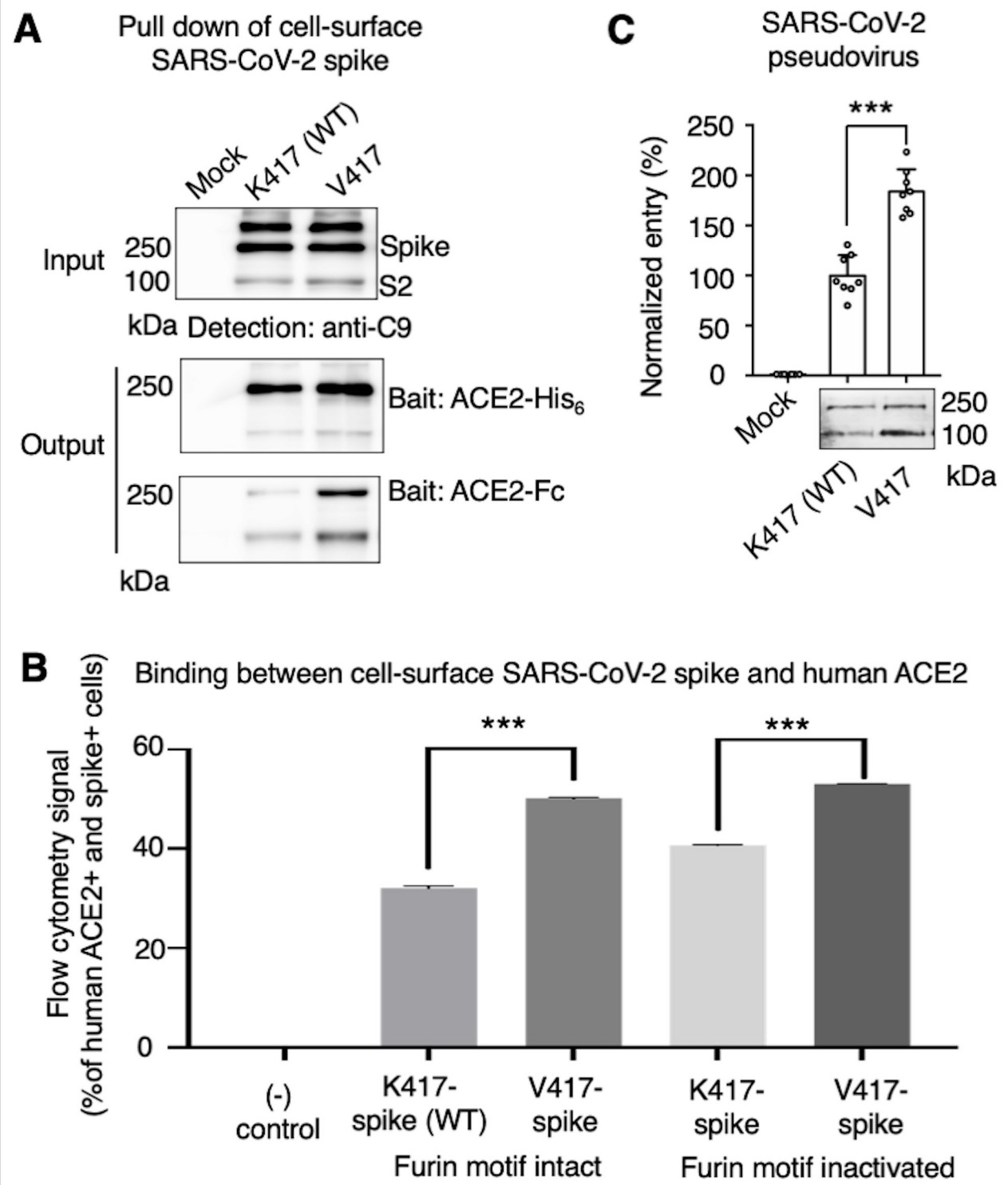

**Figure 3.** Biochemical analyses of residue 417 in regulating the conformation of membrane-anchored full-length SARS-CoV-2 spike and of the functions of the spike in different conformations. (**A**) Protein pull-down assay using recombinant human ACE2 as the bait and cell-surface-anchored full-length SARS-CoV-2 spike as the target. The spike contains either Lys417 (wild-type residue) or Val417 (mutant residue). Top: cell-surface-expressed SARS-CoV-2 spike. Middle: pull-down results using His$_6$-tagged ACE2. Bottom: pull-down results using Fc-tagged ACE2 (*Figure 3—source data 1*). The expected molecular weights of SARS-CoV-2 spike monomer and S2 monomer are ~180 kDa and ~80 kDa, respectively. (**B**) Flow cytometry assay to detect the interactions between recombinant human ACE2 and cell-surface-anchored full-length SARS-CoV-2 spike (*Figure 3—source data 2*). The spike contains either Lys417 (wild-type residue) or Val417 (mutant residue) and contains either intact furin motif or inactivated furin motif. See *Figure 3— figure supplement 1* for details of this experiment. Data are mean + SEM. A comparison (two-tailed Student's *t*-test) was performed on data between

*Figure 3 continued on next page*

*Figure 3 continued*

indicated groups (n = 3). ***p<0.001. (**C**) SARS-CoV-2 pseudovirus entry into human-ACE2-expressing cells. The virus-surface-anchored spike contains either Lys417 (wild-type residue) or Val417 (mutant residue). Top: pseudovirus entry efficiency normalized against the expression level of the spike (see bottom) (*Figure 3—source data 3*). Bottom: SARS-CoV-2 spike in packaged pseudoviruses (*Figure 3—source data 4*). Data are mean + SEM. A comparison (two-tailed Student's *t*-test) was performed on data between indicated groups (n = 8). ***p<0.001. All experiments in this figure were repeated independently three times with similar results.

The online version of this article includes the following source data and figure supplement(s) for figure 3:

**Source data 1.** Raw image for *Figure 3A*.

**Source data 2.** Numerical data for *Figure 3B*.

**Source data 3.** Numerical data for *Figure 3C*.

**Source data 4.** Raw image for *Figure 3C*.

**Figure supplement 1.** Flow cytometry assay on the interactions between cell-surface-anchored full-length trimeric spike and recombinant human ACE2.

human ACE2 using flow cytometry. The result showed that the K417V mutation reduced the RBD's binding affinity for ACE2 (*Figure 5C, Figure 5—figure supplement 1*). Thus, the K417V mutation has contrasting effects on the RBD and the trimeric spike in terms of ACE2 binding: it weakens the RBD's ability to bind to ACE2 by eliminating a favorable interaction at the RBD/ACE2 interface, but it increases the trimeric spike's probability of binding to ACE2 by enabling more spike molecules to assume the open conformation.

To summarize, we identified and investigated a molecular switch regulating the conformation of SARS-CoV-2 spike protein. We used multiple approaches: cryo-EM study of recombinant spike ecto-domains, biochemical investigation of cell-surface-anchored full-length and wild-type spikes, spike-mediated pseudovirus entry, and structural analysis of neutralizing antibody/nanobody epitopes. To

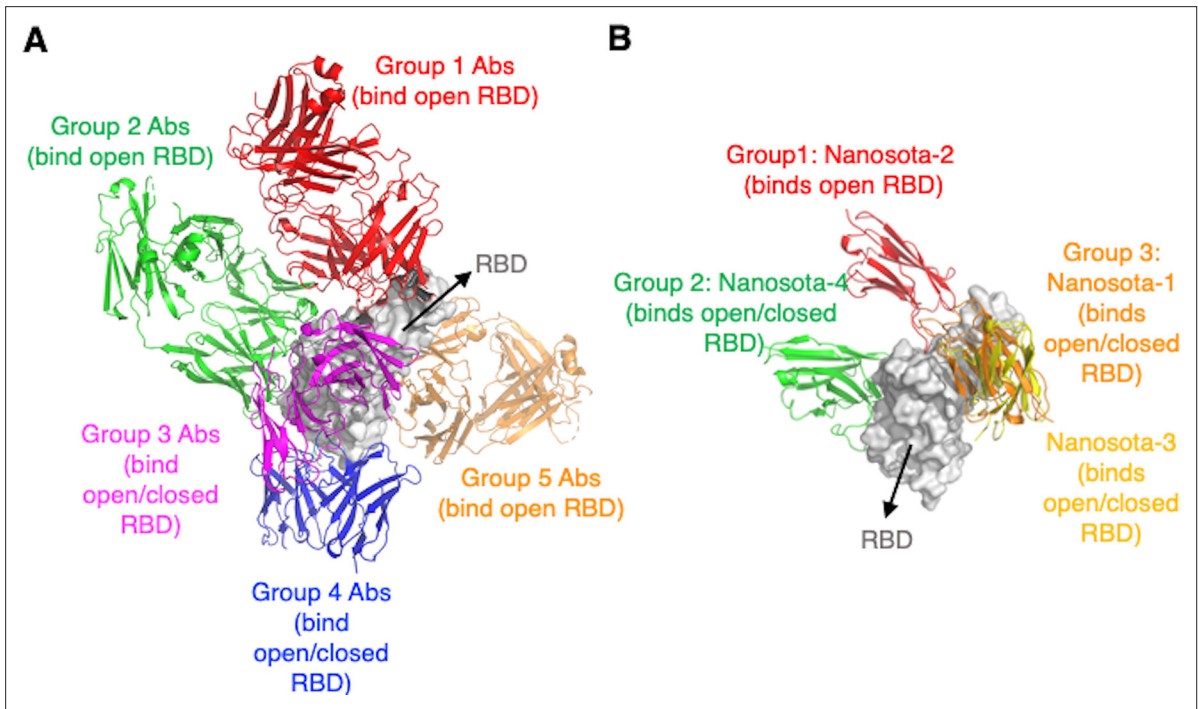

**Figure 4.** Analyses of epitopes on SARS-CoV-2 receptor-binding domain (RBD) that are accessible to neutralizing antibodies or nanobodies when the spike adopts different conformations. (**A**) Epitopes targeted by neutralizing conventional antibodies. Numerous RBD-targeting human antibodies have been discovered individually from COVID-19 patients. They bind to five groups of epitopes. Only one representative antibody for each group is shown. The PDB codes are 7N4L for group 1, 7M7W for group 2, 7M6G for group 3, 7AKD for group 4, and 7M7W for group 5. Three epitopes out of the five groups are accessible to conventional antibodies only when the spike adopts the open conformation. (**B**) Epitopes targeted by neutralizing nanobodies (single-domain antibodies). We previously discovered four RBD-targeting nanobodies from camelid animals. They bind to three groups of epitopes. The PBD codes are 7KM5 for Nanosota-1, 8G72 for Nanosota-2, 8G74 for Nanosota-3, and 8G75 for Nanosota-4. The Nanosota-2 epitope is accessible to the nanobody only when the spike adopts the open conformation.

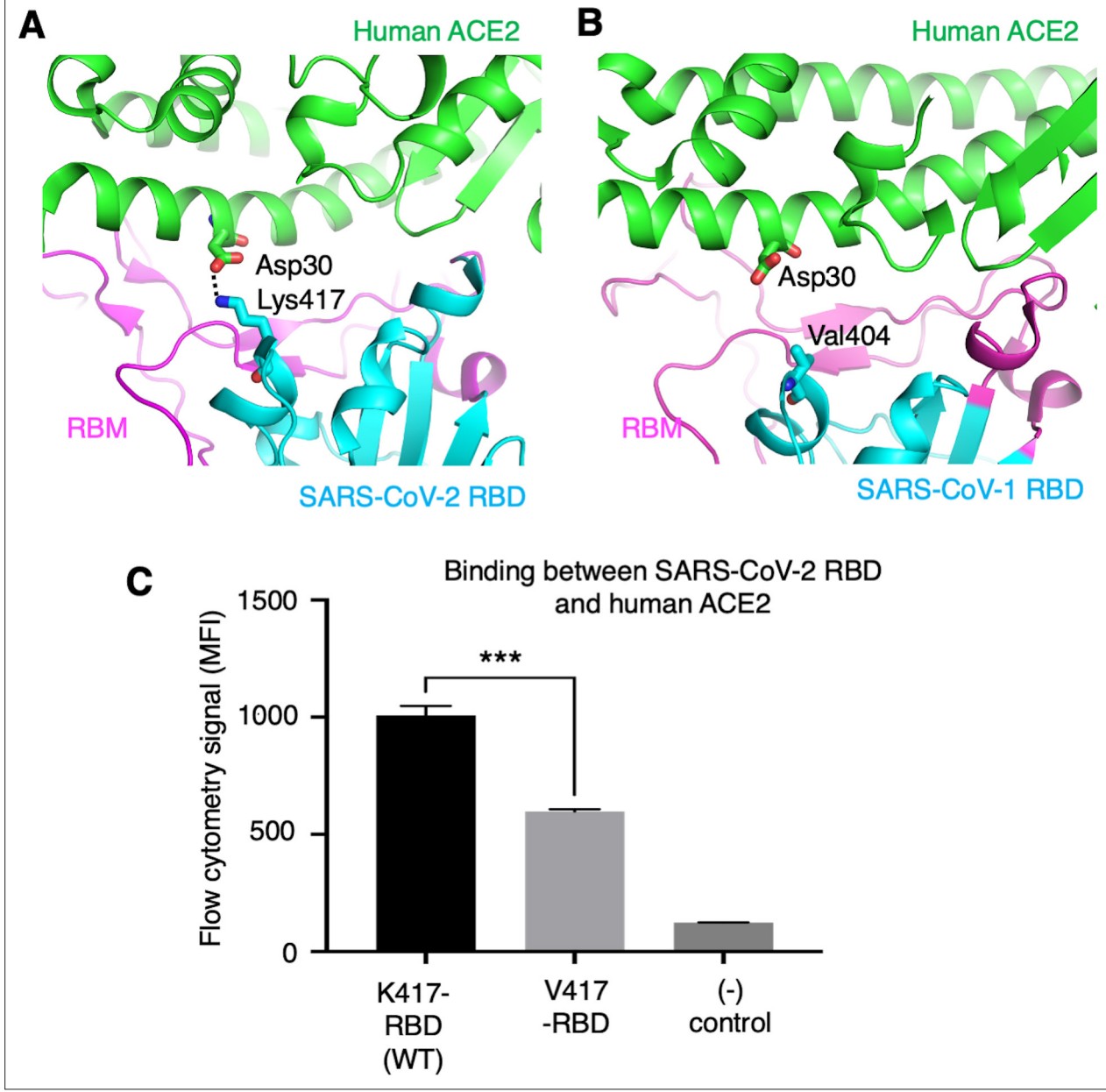

**Figure 5.** Role of residue 417 in direct interaction with ACE2 receptor. (**A**) Lys417 in SARS-CoV-2 receptor-binding domain (RBD) forms a favorable salt bridge with Asp30 in human ACE2. PDB code: 6M0J. (**B**) Val404 in SARS-CoV-1 RBD (whose position is equivalent to residue 417 in SARS-CoV-2 RBD) does not form any direct interaction with human ACE2. PDB code: 2AJF. (**C**) Flow cytometry assay to detect the interactions between recombinant SARS-CoV-2 RBD and cell-surface-anchored human ACE2 (*Figure 5—source data 1*). The RBD contains either Lys417 (wild-type residue) or Val417 (mutant residue). MFI: median fluorescence intensity. See *Figure 5—figure supplement 1* for details of this experiment. Data are mean + SEM. A comparison (two-tailed Student's *t*-test) was performed on data between indicated groups (n = 3). ***p<0.001. This experiment was repeated independently three times with similar results.

The online version of this article includes the following source data and figure supplement(s) for figure 5:

**Source data 1.** Numerical data for *Figure 5C*.

**Figure supplement 1.** Flow cytometry assay on the interactions between cell-surface-anchored human ACE2 and recombinant SARS-CoV-2 receptor-binding domain (RBD).

date, several other studies also investigated the conformation of SARS-CoV-2 spike using cryo-EM (*Wrobel et al., 2020*; *Walls et al., 2020*; *Ke et al., 2020*; *Xiong et al., 2020*; *Cai et al., 2020*; *Benton et al., 2021*), some of which gave different ratios of open and closed spikes probably due to differences in sample preparations and/or protein constructs. In this study, the two spike constructs only differ at residue 417 and the two spike samples were prepared using the same procedure. Importantly, our cryo-EM analysis is consistent with our extensive biochemical and functional approaches. These varied experimental methods complement one another, making this study one of the most thorough in examining the conformation of SARS-CoV-2 spike.

## Discussion

We previously described a unique feature of the prototypic SARS-CoV-2 spike protein: it is present as a mixture of open and closed conformations (*Shang et al., 2020b*). Building on this discovery, this study demonstrates that the conformation of SARS-CoV-2 spike is regulated, at least in part, by a structural switch. Additionally, this research sheds light on the roles of the conformational changes in the spike's functions. These findings have implications for the structure, function, and evolution of coronavirus spikes, as well as for the present and potential future coronavirus infections.

How has SARS-CoV-2 spike evolved to reach a balance of open and closed conformations? Through comparative studies of SARS-CoV-2 and SARS-CoV-1 spikes, we identified a molecular switch, RBD residue 417, that regulates the conformational changes of SARS-CoV-2 spike. Lys417 in the closed SARS-CoV-2 spike forms a hydrogen bond with another spike subunit, stabilizing the closed spike. In contrast, Val417 in SARS-CoV-1 spike cannot form such a hydrogen bond with another spike subunit, unable to stabilize the closed spike. When we introduced the K417V mutation to SARS-CoV-2 spike, more spike molecules turned open. Interestingly, Lys417 in the RBD of SARS-CoV-2 spike also directly interacts with ACE2, and we showed that the K417V mutation reduced the RBD's binding affinity for ACE2. Hence, the K417V mutation has opposing effects on the RBD and the spike for their binding to ACE2. The net outcomes of the K417 mutation are to enhance the spike's overall binding to ACE2 and the spike's capability to mediate viral entry.

Besides residue 417, there are likely other molecular factors that influence the opening and closing of SARS-CoV-2 spike. Studies have shown that N-linked glycans on the spike and fatty acids bound to the spike both play a role in regulating its conformation (*Toelzer et al., 2020*; *Sztain et al., 2021*). As for protein-based factors, a D614G mutation that emerged later in the pandemic caused SARS-CoV-2 spike to favor the open conformation (*Yurkovetskiy et al., 2020*; *Ozono et al., 2021*). Furthermore, our earlier research indicated that three lysine residues kept SARS-CoV-2 Omicron spike in the open conformation (*Ye et al., 2022*). It is worth noting that residue 417 itself has been a mutational hotspot in SARS-CoV-2 spike; in some pre-Omicron variants, it changed to either asparagine or threonine, and in most Omicron subvariants, it became asparagine (*Figure 1C*). Given that the side chains of Asn417 and Thr417 might be too short to form a direct interaction with another spike subunit, the mutations at residue 417 could have influenced the conformation of the spike in subsequent SARS-CoV-2 variants. Nevertheless, this study focuses on prototypic SARS-CoV-2, offering comprehensive structural, biochemical, and functional data that pinpoint residue 417 as a key determinant in regulating the conformation of the prototypic spike. The molecular switch at residue 417 could have been a factor in the onset of the COVID-19 pandemic, as discussed below.

How does the conformation of SARS-CoV-2 spike influence both viral entry and immune evasiveness? Our study shows that, compared to its closed form, the open spike is more effective at binding to ACE2 and facilitating viral entry. This could enhance SARS-CoV-2's ability to infect and spread among humans. We also found that certain epitopes on the RBD are accessible only in the open spike, allowing them to be targeted by neutralizing conventional antibodies and even compact single-domain nanobodies. This suggests the virus can efficiently evade hosts' immune defenses. The virus' ability to switch between open and closed spike conformations may be key to its balance of infectiousness and evasion of immunity. This delicate balance might explain the often mild symptoms and subdued, delayed immune reactions observed in SARS-CoV-2 carriers. In comparison, SARS-CoV-1 spike predominantly assumes the open conformation (89% open and 11% closed) (*Gui et al., 2017*), potentially leading to more pronounced symptoms and quicker, stronger immune reactions in those infected. Conversely, NL63-CoV spike remains closed (100% closed) (*Walls et al., 2016*), possibly resulting in even milder symptoms and more muted immune responses. Therefore, SARS-CoV-2's

blend of infectiousness and evasive abilities, differing from both the highly pathogenic SARS-CoV-1 and the milder NL63-CoV, might be a factor in its widespread transmission.

To summarize, Lys417 in SARS-CoV-2 spike aids in keeping the spike closed, potentially hindering ACE2 binding and viral entry, but providing an edge in immune evasion. Simultaneously, Lys417's direct interaction with human ACE2 boosts ACE2 binding and viral entry, countering the limitations of the closed spike. This interesting mechanism helps SARS-CoV-2 maintain a delicate balance between infectiousness and evading host immune response.

## Materials and methods

### Cell lines and plasmids

HEK293T cells (American Type Culture Collection [ATCC]) were grown in Dulbecco's modified Eagle medium (containing 10% fetal bovine serum, 2 mM L-glutamine, 100 units/mL penicillin, and 100 µg/mL streptomycin). 293F cells (Thermo Fisher) were grown in FreeStyle 293 Expression Medium (Thermo Fisher). Both mammalian cells were authenticated by ATCC using STR profiling and were also tested for mycoplasma contamination. No commonly misidentified cell lines were used.

All of the protein constructs in this study were cloned into pcDNA 3.1(+) vector (Life Technologies). Prototypic SARS-CoV-2 spike (GenBank accession number QHD43416.1) and human ACE2 (GenBank accession number NM_021804) were synthesized (GenScript Biotech) and cloned into the vector containing a C-terminal C9 tag. Prototypic SARS-CoV-2 spike ectodomain (residues 1–1211) was cloned into the vector containing either Lys417 or Val417, in addition to six mutations in S2 (F817P, A892P, A899P, A942P, K986P, V987P) (*Hsieh et al., 2020*; *Ye et al., 2022*), furin motif mutations (R682A, R683G, R685G) (*Shang et al., 2020b*), a C-terminal foldon trimerization tag, and a C-terminal His$_6$-tag. SARS-CoV-2 spike RBD (residues 319–535) was cloned into the vector containing either Lys417 or Val417, an N-terminal tPA signal peptide, and a C-terminal His$_6$-tag. Human ACE2 ectodomain (residues 1–615) was cloned into the vector containing either a C-terminal His$_6$-tag or Fc-tag.

### Comparison of residue 417 from representative SARS-CoV-2 variants

The spike sequences of representative SARS-CoV-2 variants were retrieved from https://www.who.int/activities/tracking-SARS-CoV-2-variants.

### Protein expression and purification

All of the recombinant proteins were expressed in 293F cells (Thermo Fisher) using a FreeStyle 293 mammalian cell expression system (Life Technologies) as previously described (*Wan et al., 2020b*). In brief, the His-tagged proteins were collected from cell culture medium, purified using a Ni-NTA column (Cytiva Healthcare), and purified further using a Superdex gel filtration column (Cytiva Healthcare). The Fc-tagged protein was purified in the same way as the His-tagged proteins, except that a protein A column replaced the Ni-NTA column in the procedure.

### Protein pull-down assay

Protein pull-down assay was performed using a Dynabeads immunoprecipitation kit (Invitrogen) as previously described (*Shang et al., 2020b*). Briefly, 10 µL of Dynabeads, either for His$_6$-tagged proteins or for Fc-tagged proteins, were washed with PBS buffer and then were incubated with either 8 µg ACE2-His (human ACE2 with a C-terminal His$_6$ tag) or 10 µg ACE2-Fc (human ACE2 with a C-terminal Fc tag). Subsequently, ACE2-bound beads were washed with PBS buffer plus 0.05% Tween-20 (PBST) and then were aliquoted into different tubes for later use. To prepare cell-associated SARS-CoV-2 spike (containing either Lys417 or Val417), HEK293T cells were transfected with pcDNA3.1(+) plasmid encoding the full-length spike (containing a C-terminal C9 tag). 48 hr after transfection, the spike-expressing cells were lysed in immunoprecipitation assay buffer using a sonicator and then centrifuged. The supernatants containing the solubilized cell-membrane-associated spike were incubated with the ACE2-bound beads (ACE2 was in excess of spike). The beads were then washed with PBST buffer, and the bound spike was eluted using elution buffer. The samples were then subjected to western blot analysis and detected using anti-C9-tag antibody (Thermo Fisher).

## Flow cytometry

Flow cytometry was performed to detect the interactions between recombinant human ACE2 and cell-surface-expressed full-length SARS-CoV-2 spike (containing either Lys417 or Val417 and containing either intact furin motif or inactivated furin motif). The procedure was carried out as previously described (*Maeda et al., 2022*; *Zhang et al., 2020*). Briefly, the plasmids encoding one of the C9-tagged SARS-CoV-2 spikes or vector (pcDNA3.1(+)) itself were transfected into HEK293T cells. After 36 hr, the cells were harvested and incubated with recombinant Fc-tagged human ACE2 (25 μg/mL) on ice for 1 hr. After three washes with PBS (containing 1% BSA), the cells were incubated with Alexa Fluor 488 anti-human-IgG-Fc antibody (1:500) (BioLegend Inc) and PE-labeled anti-C9-tag antibody (1:500) (Santa Cruz Biotechnology) on ice for another hour. After more washes, the cells were fixed with 4% formaldehyde. The fluorescence intensities of the cells were measured using flow cytometry (BD FACSymphony A3 cell analyzer).

Flow cytometry was also performed to detect the interactions between SARS-CoV-2 RBD (containing either Lys417 or Val417) and cell-surface-expressed human ACE2. Briefly, HEK293T cells stably expressing human ACE2 were constructed as previously described (*Shang et al., 2018*; *Geng et al., 2021*). These cells were incubated with one of the recombinant His-tagged SARS-CoV-2 RBDs (10 μg/mL) or buffer only on ice for 1 hr. After three washes with PBS (containing 1% BSA), the cells were incubated with PE-labeled anti-His-tag antibody (1:500) (Santa Cruz Biotechnology) on ice for another hour. After more washes, the cells were fixed with 4% formaldehyde, and the fluorescence intensity of the cells was measured using flow cytometry.

## Pseudovirus entry

Pseudoviruses were packaged as previously described (*Peng et al., 2017*). Briefly, pcDNA3.1(+) plasmid encoding the full-length SARS-CoV-2 spike (containing either Lys417 or Val417) was co-transfected into HEK293T cells with helper plasmid psPAX2 and reporter plasmid plenti-CMV-luc at a molar ratio of 1:1:1 using Lipofectamine 3000 (Life Technologies). The produced pseudoviruses were harvested 72 hr post transfection and then were used to enter HEK293T cells stably expressing human ACE2. After incubation at 37°C for 5 hr, medium was replaced and cells were incubated for an additional 48 hr. Cells were then washed with PBS and lysed. Aliquots of cell lysates were transferred to Optiplate-96 (PerkinElmer), followed by addition of luciferase substrate. Relative light units (RLUs) were measured using EnSpire plate reader (PerkinElmer). Meanwhile, the amounts of pseudovirus-packaged spikes were measured by western blot using anti-C9-tag antibody and then quantified using Fiji (https://imagej.net/). The RLUs were then normalized against the amounts of pseudovirus-packaged spikes.

## Western blot

Cells or pseudoviruses were mixed with SDS loading buffer and then were incubated at 95°C for 10 min. Samples were run in a 10% SDS Tris-Glycine Gel and transferred to a PVDF membrane. Anti-C9-tag or anti-His-tag monoclonal primary antibody (1:1000) (Santa Cruz Biotech) and horseradish peroxidase-conjugated mouse secondary antibody (1:10,000) (Jackson Laboratory) were used for western blotting. LAS-4000 imager (Cytiva Healthcare) was used to develop images.

## Cryo-EM grid preparation and data acquisition

The recombinant spike ectodomain (containing either Lys417 or Val417) (4 μL at 10.3–11.5 μM) was applied to freshly glow-discharged Quantifoil R1.2/1.3 300-mesh copper grids (EM Sciences), and then blotted for 4 s at 22°C under 100% chamber humidity and plunge-frozen in liquid ethane using a Vitrobot Mark IV (FEI). Cryo-EM data were collected using Latitude-S (Gatan) on a Titan Krios electron microscope (Thermo Fisher) equipped with a K3 direct electron detector with a Biocontinuum energy filter (Gatan) in counting mode at the Hormel Institute, University of Minnesota. The movies were collected at a nominal magnification of 130,000× (corresponding to 0.664 Å per pixel), a 20 eV slit width, a dose rate of 20 e- per Å$^2$ per second, and a total dose of 40 e-/Å$^2$. The statistics of cryo-EM data collection are summarized in *Supplementary file 1*.

## Image processing

Cryo-EM data were processed using cryoSPARC v4.0.3 (*Punjani et al., 2017*), and the data processing procedures are outlined in *Figure 2—figure supplements 1 and 2*. Dose-fractionated movies were

first subjected to patch motion correction with data downsampled by 4/3 (0.885333 Å/pixel after downsampling) and patch CTF estimation with MotionCor2 (*Rubinstein and Brubaker, 2015*) and CTFFIND-4.1.13 (*Rohou and Grigorieff, 2015*), respectively. Images with the defocus values outside of –0.8 to –2.4 μm or the CTF fit resolutions worse than 6 Å were excluded from further steps. Particles were picked using both Blob picker and Template picker accompanied by removing duplicate particles. Three rounds of 2D classifications were applied to remove junk particles, and good particles (57,795 or 45,999) extracted from the good 2D classes were used for ab initio reconstruction of three maps and then for the heterogeneous refinements. The good 3D class (42,970 or 29,596 particles) was finally subjected to further homogeneous, non-uniform, and CTF refinements to generate a 3.4 Å or 3.7 Å resolution map.

Particles in the good 3D classes were then imported into RELION-4.0 (*Zivanov et al., 2020*) using the csparc2star.py module (UCSF pyem v0.5. Zenodo) and subjected to signal subtraction to keep only the mixed-state receptor-binding subunit of the spike, followed by masked 3D classification. Particles in the two different classes from the masked 3D classification were then reverted to the original particles and subjected to non-uniform refinements in cryoSPARC v3.3.2. The C3 symmetry was applied for the maps with the closed state RBD. Map resolution was determined by gold-standard Fourier shell correlation (FSC) at 0.143 between the two half-maps. Local resolution variation was estimated from the two half-maps in cryoSPARC v4.0.3.

## Model building and refinement

Initial model building of the prototypic SARS-CoV-2 spike was performed in Coot-0.8.9 (*Emsley and Cowtan, 2004*) using PDBs 7TGX and 7TGY as the starting models. Several rounds of refinement in Phenix-1.16 (*Adams et al., 2010*) and manually building in Coot-0.8.9 were performed until the final reliable models were obtained. The final model has good stereochemistry as evaluated by MolProbity (*Chen et al., 2010*). The statistics of 3D reconstruction and model refinement are shown in *Supplementary file 1*. Figures were generated using UCSF Chimera X v0.93 (*Goddard et al., 2018*).

In this study, both open spikes have only one RBD standing up and two RBDs lying down. This finding aligns with the results of several other structural studies on SARS-CoV-2 spike (*Walls et al., 2020*; *Xiong et al., 2020*; *Cai et al., 2020*). However, some other studies have reported instances of open SARS-CoV-2 spike with either two or three RBDs standing up (*Benton et al., 2021*; *Yurkovetskiy et al., 2020*). The cause for this discrepancy is unclear, but could be due to different sample preparations and/or protein constructions.

## Acknowledgements

This study was supported by funding from the University of Minnesota (to FL), NIH grants R01AI089728 (to FL), R01AI157975 (to FL), U19AI171954 (to FL, BL), and R01AI110700 (to FL).

## Additional information

### Funding

| Funder | Grant reference number | Author |
| --- | --- | --- |
| National Institutes of Health | R01AI089728 | Fang Li |
| National Institutes of Health | R01AI157975 | Fang Li |
| National Institutes of Health | U19AI171954 | Fang Li |
| National Institutes of Health | R01AI110700 | Fang Li |

The funders had no role in study design, data collection and interpretation, or the decision to submit the work for publication.

## Author contributions

Qibin Geng, Yushun Wan, Conceptualization, Data curation, Formal analysis, Validation, Investigation, Visualization, Writing – review and editing; Fu-Chun Hsueh, Data curation, Validation, Investigation, Writing – review and editing; Jian Shang, Gang Ye, Data curation, Formal analysis, Validation, Investigation, Writing – review and editing; Fan Bu, Formal analysis, Validation, Visualization, Writing – review and editing; Morgan Herbst, Rowan Wilkens, Data curation, Investigation, Writing – review and editing; Bin Liu, Conceptualization, Resources, Data curation, Formal analysis, Funding acquisition, Validation, Investigation, Visualization, Writing – review and editing; Fang Li, Conceptualization, Formal analysis, Supervision, Funding acquisition, Validation, Investigation, Visualization, Writing – original draft, Project administration, Writing – review and editing

## Author ORCIDs

Gang Ye http://orcid.org/0000-0001-6034-2174
Bin Liu https://orcid.org/0000-0002-6581-780X
Fang Li https://orcid.org/0000-0002-1958-366X

## Decision letter and Author response

Decision letter https://doi.org/10.7554/eLife.74060.sa1
Author response https://doi.org/10.7554/eLife.74060.sa2

# Additional files

## Supplementary files

• Supplementary file 1. Cryo-EM data collection, refinement, and validation statistics.

• Transparent reporting form

## Data availability

All data generated or analysed during this study are included in the manuscript and supporting files. Source data files have been provided for Figures 3 and 5. The atomic models and corresponding cryo-EM density maps have been deposited into the PDB and the Electron Microscopy Data Bank, respectively, with accession numbers PDB 8UUL and EMD-42589 (prototypic SARS-CoV-2 spike containing Lys417 in the closed conformation), PDB 8UUM and EMD-42590 (prototypic SARS-CoV-2 spike containing Lys417 in the open conformation), PDB 8UUN and EMD-42591 (prototypic SARS-CoV-2 spike containing Val417 in the closed conformation), and PDB 8UUO and EMD-42592 (prototypic SARS-CoV-2 spike containing Val417 in the open conformation).

The following datasets were generated:

| Author(s) | Year | Dataset title | Dataset URL | Database and Identifier |
|---|---|---|---|---|
| Li F | 2023 | Prototypic SARS-CoV-2 spike containing Lys417 in the closed conformation | https://www.rcsb.org/structure/8UUL | RCSB Protein Data Bank, 8UUL |
| Li F | 2023 | Prototypic SARS-CoV-2 spike containing Lys417 in the closed conformation | https://www.ebi.ac.uk/emdb/EMD-42589 | Electron Microscopy Data Bank, EMD-42589 |
| Li F | 2023 | Prototypic SARS-CoV-2 spike containing Lys417 in the open conformation | https://www.rcsb.org/structure/8UUM | RCSB Protein Data Bank, 8UUM |
| Li F | 2023 | Prototypic SARS-CoV-2 spike containing Lys417 in the open conformation | https://www.ebi.ac.uk/emdb/EMD-42590 | Electron Microscopy Data Bank, EMD-42590 |
| Li F | 2023 | Prototypic SARS-CoV-2 spike containing Val417 in the closed conformation | https://www.rcsb.org/structure/8UUN | RCSB Protein Data Bank, 8UUN |

*Continued on next page*

*Continued*

| Author(s) | Year | Dataset title | Dataset URL | Database and Identifier |
|---|---|---|---|---|
| Li F | 2023 | Prototypic SARS-CoV-2 spike containing Val417 in the closed conformation | https://www.ebi.ac.uk/emdb/EMD-42591 | Electron Microscopy Data Bank, EMD-42591 |
| Li F | 2023 | Prototypic SARS-CoV-2 spike containing Val417 in the open conformation | https://www.rcsb.org/structure/8UUO | RCSB Protein Data Bank, 8UUO |
| Li F | 2023 | Prototypic SARS-CoV-2 spike containing Val417 in the open conformation | https://www.ebi.ac.uk/emdb/EMD-42592 | Electron Microscopy Data Bank, EMD-42592 |

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
