## [Editor Report]

In this valuable study, Li et al. identify a molecular switch within the SARS-CoV-1/2 spike proteins, in position 417. Based on comparisons between CoV-1 and CoV-2 sequences, the authors propose that the nature of the residue of this position determines whether the spike favors a closed or an open conformation. The authors present solid cryo-EM and biochemical data in support of their hypothesis. The paper will be of general interest to those interested in molecular mechanisms of viral infection by SARS-CoV-2 and other coronaviruses.

---

## [Decision Letter]

**Decision letter after peer review:**

Thank you for submitting your article "Molecular switches regulating the potency and immune evasiveness of SARS-CoV-2 spike protein" for consideration by *eLife*. Your article has been reviewed by 3 peer reviewers, including Lejla Zubcevic as Reviewing Editor and Reviewer #1, and the evaluation has been overseen by José Faraldo-Gómez as Senior Editor. The following individual involved in review of your submission has agreed to reveal their identity: Yi Ren (Reviewer #3).

Essential revisions:

All reviewers agreed that this is a well-executed study with significant findings. However, we agreed that a few revisions should be made prior to publication. We have compiled the list of requests, comments, and questions below.

1. In order to be able to comment on the molecular mechanism of close-to-open transition, the authors need to look at atomic models of the RBD captured in both closed and open states. The reviewers would like the authors to submit an open RBD structure. The data for this should already exist (4.6A K417V/furin motif deletion mutant 3D reconstruction presented in figure 2C). To get an idea of the general mechanism of the closed-to-open transition, the authors could build a (where appropriate, side chain-less model) of the 4.6A K417V/furin motif deletion mutant, for which the data already exists.

2. We would encourage both a comparison with the FnM-deletion (100% closed) structure and an internal comparison between the open and closed protomers in the "open" structure to gain an insight into the mechanism.

3. The way it is explained in the methods, it would appear that only the highest resolution class is taken into consideration when the percentage occupancies are calculated for the open and closed states. We are not convinced that this is the most appropriate way to analyze the data because it is inherently biased towards more structurally stable classes. Other open conformations might exist at lower resolutions. Ideally, the state occupancy percentages should be based on the total particle count. For example, assuming that the open conformation is more unstable than the closed one, the real distribution could be 90% open and 10% closed, where only the 10% are selected and contribute to the high resolution reconstruction. In this case, it'd be wrong to say that 100% of the particles are in the closed conformation when the data actually says that 100% of the high resolution particles (and 10% of the total) are in the closed conformation. At the very least, the authors should discuss this aspect of data analysis and explain the implications for their hypothesis.

4. The authors should create a figure that explains their cryo-EM workflow for analysis in detail (representative micrograph, 2D classification, 3D classifications, etc). This figure should also contain local resolution plots and Euler angle distribution plots.

5. The PDB validation and visual inspection of maps indicates that lots of residues/regions do not fit the map very well. Lots of side chains were built with no density supporting them. We encourage the authors to go back to these models and a. remove segments of the model that are not supported by experimental density and b. re-build the backbone of the protein in parts of the protein where it does not fit the density well. This is particularly an issue because the authors discuss interaction networks in regions with poor density (Lys417 and it's interaction partner are part of this region too). As requested in point 4, a local resolution plot of the 3D model will also come in handy for the readers here to easily estimate the accuracy of the map in different regions of the protein.

6. The authors do not show a WT control in their mouse immunization experiments (figure 5). This control should be included.

7. The authors should comment on why we only observe one RBD in the up conformation, despite point mutations, especially Lys417Ala, being introduced to all 3 protomers? Are there factors that might be missing from the experimental system that might play a role in stabilizing the "fully open" i.e. all 3 RBDs in the up conformation.

8. Is there co-operativity in the trimer? E.g., does releasing one RBD change the open-to-closed equilibrium for the other two? And do the furin motif and the salt bridge act independently of each other?

9. The WT SARS-CoV-2 (containing the FnM) still performs better than the point mutant (FnM-point) in virus entry assays, suggesting that the charges within the FnM might play an important role. Have the authors created a SARS-CoV-2 with a GSGS-linker for comparison? And SARS-CoV-2 with the Arg replaced with Lys (which wouldn't be cleaved by furin)?

10. Is it accurate that the structure of RaTG13-CoV is always in the closed conformation or is this an artefact of experimental conditions? This structure was crosslinked, which could have led to a 100% closed population.

11. Is it known what allows the RaTG13 spike molecule to switch into an open conformation? Are charged residues expected at play in this scenario as well (i.e., does RaTG13 have a lysine residue in the same position as SARS-CoV-2)? For comparison purposes, it would be helpful to know what residue is at the Lys417-equivalent position in RaTG13.

12. How well does the WT SARS-CoV-1 spike construct (controlled for pre-fusion) bind ACE2 and behave in the pseudovirus entry experiment? Is it comparable to the SARS-CoV-2 K417V mutant? If it is, this would suggest that the two major switches for dictating the open/closed confirmations are predominantly due to the presence of the furin motif loop and the salt bridge. If not, this could point to other factors involved in contributing to the conformational differences between SARS-CoV-1 and -2.

13. Lys417 has been identified as important for ACE2 binding. Can the authors comment on this, in the light of this new data.

14. Glycans and fatty acids have also been suggested to play a role in the open-to-closed transition. Can the authors comment on their potential roles in the light of this new data.

15. The illustration of the mechanism of the closed-to-open transition Figure S3A was difficult to interpret. Similarly, the interfaces between S1 and NTD were also difficult to glean from the figure. We would encourage the authors to make two separate figures (or at least, two panels in one figure) to illustrate this better. Also, a surface representation might work better to show the interface between S1 and NTD.

16. Could the authors include more spike structures (i.e. closed form of D614G) for comparing the interfaces in Figure S3? Spike proteins features extensive conformational heterogeneity. Even within the same category of open or closed spike, further classification generates slightly different structures. It is not entirely clear what level of changes is significant.

17. Line 99-101: The authors should quote A. C. Walls et al., Structure, Function, and Antigenicity of the SARS-CoV-2 Spike Glycoprotein. Cell, (2020).

18. Line 136-137: Reference for the double proline/C-term foldon.

19. It would be helpful to include expected molecular masses in the legends of Figures 1C, 3B and 4B where expected bands should appear.

20. It would be very helpful to include a table of experimental outcomes that includes the protein conformations of all WT proteins (CoV-1, CoV-2, and RaTG13) and mutants, a summary of ACE2 binding and pseudovirus entry.

21. In line 441, is "26,126 particles" correct? This number seems to refer to particles selected from 2D for the entire dataset.

*Reviewer #1 (Recommendations for the authors):*

The conformation of the SARS-CoV-2 spike protein receptor binding domain (RBD) has implications for binding to the host cell surface receptor: the open conformation is required for binding, but it also makes the virus more likely to be detected by the host's immune system. SARS-CoV-2 spike protein RBD has been observed with equal probability in both open and closed conformations but the mechanism of the transition from the closed to the open state is not known.

Here, Wan et al. present data that suggests that the flexible linker between S1 and S2, which contains a furin binding motif, is an important factor in facilitating this transition.

Importantly, the authors' findings suggest that it is not the furin motif itself but the flexible linker that facilitates the transition to the open conformation.

The study stretches from basic biochemistry to structural biology to in vivo experiments. However, the work can be strengthened by some additional data, data analysis and improved presentation of the structural biology section. In addition, some clarification regarding the in vivo data is required.

The authors present an interesting hypothesis that by being equally able to occupy the open and closed RBD conformations, SARS-Cov-2 has evolved to have high viral potency as well as successfully evade the host immune response. This is an interesting insight into viral evolution which may lead to novel avenues for future research.

I have the following comments:

1. To be able to comment on the molecular mechanism of close-to-open transition, the authors need to look at atomic models of the RBD captured in both closed and open states. The reviewers would like the authors to submit an open RBD structure. The data for this should already exist (4.6A K417V/furin motif deletion mutant 3D reconstruction presented in figure 2C). To get an idea of the general mechanism of the closed-to-open transition, the authors could build a (where appropriate, side chain-less model) of the 4.6A K417V/furin motif deletion mutant, for which the data already exists.

2. The authors should compare the open RBD structure with the FnM-deletion (100% closed) structure. In addition, they should make an internal comparison between the open and closed protomers in the "open" structure to gain an insight into the mechanism.

3. The way it is explained in the methods, only the highest resolution class is taken into consideration when the percentage occupancies are calculated for the open and closed states. This approach is inherently biased towards more structurally stable classes. Other open conformations might exist at lower resolutions. Ideally, the state occupancy percentages should be based on the total particle count.

For example, assuming that the open conformation is more unstable than the closed one, the real distribution could be 90% open and 10% closed, where only the 10% are selected and contribute to the high-resolution reconstruction. In this case, it'd be wrong to say that 100% of the particles are in the closed conformation when the data actually says that 100% of the high-resolution particles (and 10% of the total) are in the closed conformation. At the very least, the authors should discuss this aspect of data analysis and explain the implications for their hypothesis.

4. The authors should create a figure that explains their cryo-EM workflow for analysis in detail (representative micrograph, 2D classification, 3D classifications, etc). This figure should also contain local resolution plots and Euler angle distribution plots.

5. The PDB validation and visual inspection of maps indicates that lots of residues/regions do not fit the map very well. Lots of side chains were built with no density supporting them. The authors should: a. remove segments of the model that are not supported by experimental density and b. re-build the backbone of the protein in parts of the protein where it does not fit the density well. This is particularly an issue because the authors discuss interaction networks in regions with poor density (Lys417 and its interaction partner are part of this region too).

6. The authors do not show a WT control in their mouse immunization experiments (figure 5). This control should be included.

7. The authors should discuss the Lys417 residue and its involvement in ACE2 binding.*Reviewer #2 (Recommendations for the authors):*

Wan et al., investigate some of the major structural differences between the SARS-CoV-1, SARS-CoV-2, and RaTG13-CoV spike proteins to better understand why SARS-CoV-2 led to a global pandemic in humans. In particular, the study focuses on the SARS-CoV-2 spike (S) protein in the pre-fusion conformation, used by the virus to bind and enter host cells. The pre-fusion conformation of the S protein is known to have either open, closed or a mixture of open/closed conformations within the receptor-binding domain (RBD), depending on the type of coronavirus. The SARS-CoV-2 RBD is found in both the open and closed conformations while SARS-CoV-1 and RaTG13-CoV are found in the open and closed, respectively. The authors investigate the contributions of two unique structural features of the SARS-CoV-2 S protein, a furin binding motif within a loop as well as a lysine residue (position 417), to understand the observed mixture of conformations. Using a combination of mutational analyses, biochemical assays, structural studies, and mouse models, the authors show that both structural motifs (the furin loop and lysine residue) contribute to the ability of the protein to switch between the open and closed conformations. Furthermore, this switching between conformations most likely contributes to the effectiveness of SARS-CoV-2 establishing widespread infections in humans. Overall, this very interesting study is well conducted and will aid in the development of additional SARS-CoV-2 studies, as well as inform how other coronaviruses may employ the same strategies.

Comments:

1) It would be important to know how well the WT SARS-CoV-1 spike construct (controlled for pre-fusion) binds ACE2 and behaves in the pseudovirus entry experiment. Is it comparable to the SARS-CoV-2 K417V mutant? If it is, this would suggest that the two major switches for dictating the open/closed confirmations are predominantly due to the presence of the furin motif loop and the salt bridge. If not, this could point to other factors involved in contributing to the conformational differences between SARS-CoV-1 and -2.*Reviewer #3 (Recommendations for the authors):*

In this work, the authors have aimed to identify molecular switches regulating the conformations of SARS-CoV-2 spike protein and to investigate their effects on the potency and immune evasiveness of SARS-CoV-2. The authors have identified two such molecular switches, the furin motif and the K417 residue at the interface of the RBD domains in closed spike. The strength of the work is the comprehensive approach (SARS-CoV-2/RaTG13-CoV engineering, ACE2 binding, cryo-EM, pseudovius entry assays, etc). The authors clearly demonstrate the effects of furin motif and K417 on the ratios of open and closed spikes, and consistently, on ACE2 binding and pseudovirus entry. The correlation between spike conformations and immune evasiveness is an intriguing observation, although it remains to be more vigorously tested in future studies.

[Editors’ note: further revisions were suggested prior to acceptance, as described below.]

Thank you for resubmitting your work entitled "Lys417 acts as a molecular switch that regulates the conformation of SARS-CoV-2 spike protein" for further consideration by *eLife*. Your revised article has been evaluated by Diane Harper (Senior Editor) and a Reviewing Editor.

The manuscript has been improved but there are some remaining issues that need to be addressed, as outlined below:

*Reviewer #1 (Recommendations for the authors):*

Here, Li et al. identify a molecular switch within the SARS-CoV-1/2 spike proteins, in position 417. Based on comparisons between CoV-1 and CoV-2 sequences, the authors propose that the nature of the residue of this position determines whether the spike favors a closed or an open conformation. The authors present cryo-EM and biochemical data in support of their hypothesis. The paper will be of general interest to those interested in molecular mechanisms of viral infection by SARS-CoV-2 and other coronaviruses.

The authors have submitted a streamlined and improved manuscript. I only have a few comments to add.

– My main concern is that the word "affinity" is not used properly in some parts of the paper. For example, in the abstract, the authors state "The net outcomes of this (K417V) mutation are to allow the spike to bind ACE2 more tightly…". This is not really what the data says- the mutation seems to increase the probability of "open" spike and therefore increases the probability of spike binding. In fact, the data shows that K417V binds less tightly to ACE2 than K417.

In lines 116-118 the authors write "It has been shown that ACE2 only binds to the standing-up RBD in the open spike (28, 33), suggesting that open spike binds to ACE2 with higher affinity than the closed spike." A more accurate statement would be that the open spike conformation is necessary for ACE2 binding.

This continues in lines 127-129 "The pull-down result showed that compared to K417-spike, more V417-spike molecules had been pulled down by ACE2, suggesting that V417-spike has a higher affinity for ACE2 (Figure 3A)." This experiment does not say anything about binding affinity, but it does suggest that the binding probability between the spike protein and ACE2 is increased in V417.

In the summary of findings in lines 184-187: "it weakens the RBD's ability to bind to ACE2 by eliminating a favorable interaction at the RBD/ACE2 interface, but it strengthens the trimeric spike's binding affinity for ACE2 by enabling more spike molecules to assume the open conformation." A more accurate statement would be that the V417 increases the probability of binding to ACE2 because it increases the probability of the open spike conformation.

---

## [Author Response]

Essential revisions:All reviewers agreed that this is a well-executed study with significant findings. However, we agreed that a few revisions should be made prior to publication. We have compiled the list of requests, comments, and questions below.1. In order to be able to comment on the molecular mechanism of close-to-open transition, the authors need to look at atomic models of the RBD captured in both closed and open states. The reviewers would like the authors to submit an open RBD structure. The data for this should already exist (4.6A K417V/furin motif deletion mutant 3D reconstruction presented in figure 2C). To get an idea of the general mechanism of the closed-to-open transition, the authors could build a (where appropriate, side chain-less model) of the 4.6A K417V/furin motif deletion mutant, for which the data already exists.

The revised manuscript centers on residue 417, which acts as a molecular switch that regulates the conformation of SARS-CoV-2 spike. It has omitted the data and discussion about the furin motif deletion mutant (Figure 1). Accordingly, the title of the manuscript has been changed to “Lys417 acts as a molecular switch regulating the conformation of SARS-CoV-2 spike protein”.

In the revised manuscript, we have developed new stabilized versions of SARS-CoV-2 spike ectodomain. This was achieved by incorporating six proline mutations into the S2 subunit, an increase from the two proline mutations used in earlier versions. Subsequently, we acquired new cryo-EM data for these proteins at significantly improved resolutions and constructed new atomic models for each of the samples (Figure 2).

2. We would encourage both a comparison with the FnM-deletion (100% closed) structure and an internal comparison between the open and closed protomers in the "open" structure to gain an insight into the mechanism.

The revised manuscript centers on residue 417, which acts as a molecular switch that regulates the conformation of SARS-CoV-2 spike. It has omitted the data and discussion about the furin motif deletion mutant (Figure 1). Accordingly, the title of the manuscript has been changed to “Lys417 acts as a molecular switch regulating the conformation of SARS-CoV-2 spike protein”.

3. The way it is explained in the methods, it would appear that only the highest resolution class is taken into consideration when the percentage occupancies are calculated for the open and closed states. We are not convinced that this is the most appropriate way to analyze the data because it is inherently biased towards more structurally stable classes. Other open conformations might exist at lower resolutions. Ideally, the state occupancy percentages should be based on the total particle count. For example, assuming that the open conformation is more unstable than the closed one, the real distribution could be 90% open and 10% closed, where only the 10% are selected and contribute to the high resolution reconstruction. In this case, it'd be wrong to say that 100% of the particles are in the closed conformation when the data actually says that 100% of the high resolution particles (and 10% of the total) are in the closed conformation. At the very least, the authors should discuss this aspect of data analysis and explain the implications for their hypothesis.

Please see Figure 2—figure supplement1, Figure 2—figure supplement2, and Supplementary File 1 for the updated cryo-EM procedures.

4. The authors should create a figure that explains their cryo-EM workflow for analysis in detail (representative micrograph, 2D classification, 3D classifications, etc). This figure should also contain local resolution plots and Euler angle distribution plots.

Please see Figure 2—figure supplement1 and Figure 2—figure supplement2 for the cryo-EM workflow.

5. The PDB validation and visual inspection of maps indicates that lots of residues/regions do not fit the map very well. Lots of side chains were built with no density supporting them. We encourage the authors to go back to these models and a. remove segments of the model that are not supported by experimental density and b. re-build the backbone of the protein in parts of the protein where it does not fit the density well. This is particularly an issue because the authors discuss interaction networks in regions with poor density (Lys417 and it's interaction partner are part of this region too). As requested in point 4, a local resolution plot of the 3D model will also come in handy for the readers here to easily estimate the accuracy of the map in different regions of the protein.

The cryo-EM densities and the corresponding atomic models presented in this revision represent a substantial improvement over our previous submission. For detailed comparisons and data, refer to Figure 2—figure supplement1, Figure 2—figure supplement2, and Supplementary File 1.

6. The authors do not show a WT control in their mouse immunization experiments (figure 5). This control should be included.

The revised manuscript no longer includes mouse immunization data. Instead, it presents findings that the closed spike can evade the binding of many neutralizing antibodies and nanobodies (Figure 4).

7. The authors should comment on why we only observe one RBD in the up conformation, despite point mutations, especially Lys417Ala, being introduced to all 3 protomers? Are there factors that might be missing from the experimental system that might play a role in stabilizing the "fully open" i.e. all 3 RBDs in the up conformation.

We have added the following comment to the revised manuscript:

“In this study, both of the open spikes have only one RBD standing up and two RBDs lying down. This finding aligns with the results of several other structural studies on SARS-CoV-2 spike (*29, 40, 41*). However, some other studies have reported instances of open SARS-CoV-2 spike with either two or three RBDs standing up (*42, 45*). The cause for this discrepancy is unclear, but could be due to different sample preparations and/or protein constructions.”

8. Is there co-operativity in the trimer? E.g., does releasing one RBD change the open-to-closed equilibrium for the other two? And do the furin motif and the salt bridge act independently of each other?

We don't have the answers to the first two questions, and we believe they fall outside the scope of our current study. Regarding the last question, we have excluded data and discussions about the furin deletion mutant spike from the revised manuscript. Instead, the focus has shifted to residue 417 in the revised manuscript, pinpointing it as an important determinant that regulates the conformation of SARS-CoV-2 spike.

9. The WT SARS-CoV-2 (containing the FnM) still performs better than the point mutant (FnM-point) in virus entry assays, suggesting that the charges within the FnM might play an important role. Have the authors created a SARS-CoV-2 with a GSGS-linker for comparison? And SARS-CoV-2 with the Arg replaced with Lys (which wouldn't be cleaved by furin)?

We have excluded data and discussions about the furin deletion mutant spike from the revised manuscript. Instead, the focus has shifted to residue 417 in the revised manuscript, pinpointing it as an important determinant that regulates the conformation of SARS-CoV-2 spike.

10. Is it accurate that the structure of RaTG13-CoV is always in the closed conformation or is this an artefact of experimental conditions? This structure was crosslinked, which could have led to a 100% closed population.

The revised manuscript no longer includes discussion regarding the RaTG13-CoV spike.

11. Is it known what allows the RaTG13 spike molecule to switch into an open conformation? Are charged residues expected at play in this scenario as well (i.e., does RaTG13 have a lysine residue in the same position as SARS-CoV-2)? For comparison purposes, it would be helpful to know what residue is at the Lys417-equivalent position in RaTG13.

The revised manuscript no longer includes discussion regarding the RaTG13-CoV spike.

12. How well does the WT SARS-CoV-1 spike construct (controlled for pre-fusion) bind ACE2 and behave in the pseudovirus entry experiment? Is it comparable to the SARS-CoV-2 K417V mutant? If it is, this would suggest that the two major switches for dictating the open/closed confirmations are predominantly due to the presence of the furin motif loop and the salt bridge. If not, this could point to other factors involved in contributing to the conformational differences between SARS-CoV-1 and -2.

In response to the first question, the functions of the SARS-CoV-1 and SARS-CoV-2 spikes were thoroughly compared in one of our earlier publications (PubMed ID 32376634). The present study is specifically concentrated on SARS-CoV-2 spike.

In the revised manuscript, we added the percentages of open and closed SARS-CoV-1 spike particles:

“SARS-CoV-1 spike predominantly assumes an open conformation (89% open and 11% closed) (*28*).”

We also noted that there may be other factors contributing to the conformations of coronavirus spikes:

“Besides residue 417, there are likely other molecular factors that influence the opening and closing of SARS-CoV-2 spike. Studies have shown that N-linked glycans on the spike and fatty acids bound to the spike both play a role in regulating its conformation (*43, 44*). As for protein-based factors, a D614G mutation that emerged later in the pandemic caused SARS-CoV-2 spike to favor the open conformation (*45, 46*). Furthermore, our earlier research indicated that three lysine residues kept SARS-CoV-2 Omicron spike in the open conformation (*32*).”

13. Lys417 has been identified as important for ACE2 binding. Can the authors comment on this, in the light of this new data.

The revised manuscript includes new data indicating that the K417V mutation decreases the direct binding of the RBD to human ACE2 (Figure 5). Thus, our study reveals that the K417V mutation has opposing effects on the spike’s function: it opens up the spike for better ACE2 binding while weakening the RBD's direct binding to ACE2.

14. Glycans and fatty acids have also been suggested to play a role in the open-to-closed transition. Can the authors comment on their potential roles in the light of this new data.

We have added the following discussion to the revised manuscript:

“Besides residue 417, there are likely other molecular factors that influence the opening and closing of SARS-CoV-2 spike. Studies have shown that N-linked glycans on the spike and fatty acids bound to the spike both play a role in regulating its conformation (*43, 44*).”

15. The illustration of the mechanism of the closed-to-open transition Figure S3A was difficult to interpret. Similarly, the interfaces between S1 and NTD were also difficult to glean from the figure. We would encourage the authors to make two separate figures (or at least, two panels in one figure) to illustrate this better. Also, a surface representation might work better to show the interface between S1 and NTD.

The revised manuscript no longer includes the packing analysis of the interfaces. Due to the relative mobility of the RBDs, an accurate packing analysis would require cryo-EM data of even higher resolutions. Therefore, the revised manuscript concentrates on insights gleaned from the overall cryo-EM structures, which are supported by a variety of biochemical and functional data. Collectively, our results reveal that residue 471 is the key regulator of SARS-CoV-2 spike's conformation.

16. Could the authors include more spike structures (i.e. closed form of D614G) for comparing the interfaces in Figure S3? Spike proteins features extensive conformational heterogeneity. Even within the same category of open or closed spike, further classification generates slightly different structures. It is not entirely clear what level of changes is significant.

The revised manuscript no longer includes the packing analysis of the interfaces. Due to the relative mobility of the RBDs, an accurate packing analysis would require cryo-EM data of even higher resolutions. Therefore, the revised manuscript concentrates on insights gleaned from the overall cryo-EM structures, which are supported by a variety of biochemical and functional data. Collectively, our results reveal that residue 471 is the key regulator of SARS-CoV-2 spike's conformation.

17. Line 99-101: The authors should quote A. C. Walls et al., Structure, Function, and Antigenicity of the SARS-CoV-2 Spike Glycoprotein. Cell, (2020).

This citation has been added to the revised manuscript:

“We identified residue 417 as potentially a key difference between the two spikes: in the closed SARS-CoV-2 trimeric spike, Lys417 in the RBD from one spike subunit forms a hydrogen bond with the main chain of Asn370 in the RBD from another spike subunit, stabilizing the RBDs in the closed conformation (29) (Figure 1B).”

18. Line 136-137: Reference for the double proline/C-term foldon.

References have been added to the revised manuscript:

“To stabilize the recombinant spike ectodomain, we introduced six proline mutations into its S2 subunit to lock up its pre-fusion structure, introduced mutations to its furin motif to prevent it from being cleaved during molecular maturation, and added a C-terminal foldon tag to facilitate its trimerization (20, 31, 32).”

19. It would be helpful to include expected molecular masses in the legends of Figures 1C, 3B and 4B where expected bands should appear.

In the revised figure 2 legend, we added the following description:

“The expected molecular weights of SARS-CoV-2 spike monomer and S2 monomer are ~180 kDa and ~80 kDa, respectively.”

20. It would be very helpful to include a table of experimental outcomes that includes the protein conformations of all WT proteins (CoV-1, CoV-2, and RaTG13) and mutants, a summary of ACE2 binding and pseudovirus entry.

In the revised manuscript, we have omitted the supplementary table that cataloged the conformation of spikes from multiple coronaviruses, including RaTG13. We believe that such an extensive summary of the literature is more suited to a comprehensive review article. Instead, the revised manuscript narrows its comparison to SARS-CoV-2, SARS-CoV-1, and NL63-CoV, as all three of these viruses use ACE2 as their receptor but result in different patient symptoms. We have encapsulated the conformations of the other two spikes as follows:

“SARS-CoV-1 spike predominantly assumes the open conformation (89% open and 11% closed) (28).” “ NL63-CoV spike remains closed (100% closed) (26).”

We also note that:

“To date, several other studies also investigated the conformation of SARS-CoV-2 spike using cryo-EM (27, 29, 39-42), some of which gave different ratios of open and closed spikes probably due to differences in sample preparations and/or protein constructions. In this study, the two spike constructs only differ at residue 417 and the two spike samples were prepared using the same procedure. Importantly, our cryo-EM analysis is consistent with our extensive biochemical and functional approaches. These varied experimental methods complement one another, making this study one of the most thorough in examining the conformation of SARS-CoV-2 spike.”

21. In line 441, is "26,126 particles" correct? This number seems to refer to particles selected from 2D for the entire dataset.

Please see Figure 2—figure supplement1, Figure 2—figure supplement2, and Supplementary File 1 for the updated cryo-EM procedures.

Reviewer #1 (Recommendations for the authors):1. To be able to comment on the molecular mechanism of close-to-open transition, the authors need to look at atomic models of the RBD captured in both closed and open states. The reviewers would like the authors to submit an open RBD structure. The data for this should already exist (4.6A K417V/furin motif deletion mutant 3D reconstruction presented in figure 2C). To get an idea of the general mechanism of the closed-to-open transition, the authors could build a (where appropriate, side chain-less model) of the 4.6A K417V/furin motif deletion mutant, for which the data already exists.

The revised manuscript centers on residue 417, which acts as a molecular switch that regulates the conformation of SARS-CoV-2 spike. It has omitted the data and discussion about the furin motif deletion mutant (Figure 1). Accordingly, the title of the manuscript has been changed to “Lys417 acts as a molecular switch regulating the conformation of SARS-CoV-2 spike protein”.

2. The authors should compare the open RBD structure with the FnM-deletion (100% closed) structure. In addition, they should make an internal comparison between the open and closed protomers in the "open" structure to gain an insight into the mechanism.

The revised manuscript centers on residue 417, which acts as a molecular switch that regulates the conformation of SARS-CoV-2 spike. It has omitted the data and discussion about the furin motif deletion mutant (Figure 1). Accordingly, the title of the manuscript has been changed to “Lys417 acts as a molecular switch regulating the conformation of SARS-CoV-2 spike protein”.

3. The way it is explained in the methods, only the highest resolution class is taken into consideration when the percentage occupancies are calculated for the open and closed states. This approach is inherently biased towards more structurally stable classes. Other open conformations might exist at lower resolutions. Ideally, the state occupancy percentages should be based on the total particle count.For example, assuming that the open conformation is more unstable than the closed one, the real distribution could be 90% open and 10% closed, where only the 10% are selected and contribute to the high-resolution reconstruction. In this case, it'd be wrong to say that 100% of the particles are in the closed conformation when the data actually says that 100% of the high-resolution particles (and 10% of the total) are in the closed conformation. At the very least, the authors should discuss this aspect of data analysis and explain the implications for their hypothesis.

Please see Figure 2—figure supplement1, Figure 2—figure supplement2, and Supplementary File 1 for the updated cryo-EM procedures.

4. The authors should create a figure that explains their cryo-EM workflow for analysis in detail (representative micrograph, 2D classification, 3D classifications, etc). This figure should also contain local resolution plots and Euler angle distribution plots.

Please see Figure 2—figure supplement1 and Figure 2—figure supplement2 for the cryo-EM workflow.

5. The PDB validation and visual inspection of maps indicates that lots of residues/regions do not fit the map very well. Lots of side chains were built with no density supporting them. The authors should: a. remove segments of the model that are not supported by experimental density and b. re-build the backbone of the protein in parts of the protein where it does not fit the density well. This is particularly an issue because the authors discuss interaction networks in regions with poor density (Lys417 and its interaction partner are part of this region too).

The cryo-EM densities and the corresponding atomic models presented in this revision represent a substantial improvement over our previous submission. For detailed comparisons and data, refer to Figure 2—figure supplement1, Figure 2—figure supplement2, and Supplementary File 1.

6. The authors do not show a WT control in their mouse immunization experiments (figure 5). This control should be included.

The revised manuscript no longer includes mouse immunization data. Instead, it presents findings that the closed spike can evade the binding of many neutralizing antibodies and nanobodies (Figure 4).

7. The authors should discuss the Lys417 residue and its involvement in ACE2 binding.

The revised manuscript includes new data indicating that the K417V mutation decreases the direct binding of the RBD to human ACE2 (Figure 5). Thus, our study reveals that the K417V mutation has opposing effects on the spike’s function: it opens up the spike for better ACE2 binding while weakening the RBD's direct binding to ACE2.

Reviewer #2 (Recommendations for the authors):Comments:1) It would be important to know how well the WT SARS-CoV-1 spike construct (controlled for pre-fusion) binds ACE2 and behaves in the pseudovirus entry experiment. Is it comparable to the SARS-CoV-2 K417V mutant? If it is, this would suggest that the two major switches for dictating the open/closed confirmations are predominantly due to the presence of the furin motif loop and the salt bridge. If not, this could point to other factors involved in contributing to the conformational differences between SARS-CoV-1 and -2.

This revised manuscript focuses on residue 417 of SARS-CoV-2 spike. The comparison between the functions of SARS-CoV-1 and SARS-CoV-2 spikes was described in detail in one of our earlier publications (PubMed ID 32376634).

[Editors’ note: what follows is the authors’ response to the second round of review.]

Reviewer #1 (Recommendations for the authors):The authors have submitted a streamlined and improved manuscript. I only have a few comments to add.– My main concern is that the word "affinity" is not used properly in some parts of the paper. For example, in the abstract, the authors state "The net outcomes of this (K417V) mutation are to allow the spike to bind ACE2 more tightly…". This is not really what the data says- the mutation seems to increase the probability of "open" spike and therefore increases the probability of spike binding. In fact, the data shows that K417V binds less tightly to ACE2 than K417.

In the revised manuscript, we have changed “allow the spike to bind ACE2 more tightly” to “allow the spike to bind ACE2 with higher probability”.

In lines 116-118 the authors write "It has been shown that ACE2 only binds to the standing-up RBD in the open spike (28, 33), suggesting that open spike binds to ACE2 with higher affinity than the closed spike." A more accurate statement would be that the open spike conformation is necessary for ACE2 binding.

In the revised manuscript, we have changed “suggesting that open spike binds to ACE2 with higher affinity than does the closed spike” to “suggesting that the open spike conformation is necessary for ACE2 binding.”

This continues in lines 127-129 "The pull-down result showed that compared to K417-spike, more V417-spike molecules had been pulled down by ACE2, suggesting that V417-spike has a higher affinity for ACE2 (Figure 3A)." This experiment does not say anything about binding affinity, but it does suggest that the binding probability between the spike protein and ACE2 is increased in V417.

In the revised manuscript, we have changed “suggesting that V417-spike has a higher affinity for ACE2” to “suggesting that V417-spike had an increased probability of binding to ACE2”.

In the summary of findings in lines 184-187: "it weakens the RBD's ability to bind to ACE2 by eliminating a favorable interaction at the RBD/ACE2 interface, but it strengthens the trimeric spike's binding affinity for ACE2 by enabling more spike molecules to assume the open conformation." A more accurate statement would be that the V417 increases the probability of binding to ACE2 because it increases the probability of the open spike conformation.

In the revised manuscript, we have changed “it strengthens the trimeric spike's binding affinity for ACE2” to “it increases the trimeric spike's probability of binding to ACE2”.